# Physicochemical and In Vitro Digestion Properties of Curcumin-Loaded Solid Lipid Nanoparticles with Different Solid Lipids and Emulsifiers

**DOI:** 10.3390/foods12102045

**Published:** 2023-05-18

**Authors:** Yasi Yu, Dechu Chen, Yee Ying Lee, Nannan Chen, Yong Wang, Chaoying Qiu

**Affiliations:** 1JNU-UPM International Joint Laboratory on Plant Oil Processing and Safety, Department of Food Science and Engineering, Jinan University, Guangzhou 510632, China; yuyasi0220@163.com (Y.Y.); dechu_chen@163.com (D.C.); twyong@jnu.edu.cn (Y.W.); 2Guangdong International Joint Research Center for Oilseed Biorefinery, Nutrition and Safety, Guangzhou 510632, China; 3School of Science, Monash University Malaysia, Bandar Sunway 47500, Selangor, Malaysia; lee.yeeying@monash.edu; 4Department of Nutrition and Food Hygiene, Guangzhou Medical University, Guangzhou 511436, China; nnchen@gzhmu.edu.cn

**Keywords:** curcumin, solid lipid nanoparticles, surfactants, medium- and long chain diacylglycerol, in vitro release, bioavailability

## Abstract

Curcumin-loaded solid lipid nanoparticles (Cur-SLN) were prepared using medium- and long chain diacylglycerol (MLCD) or glycerol tripalmitate (TP) as lipid matrix and three kinds of surfactants including Tween 20 (T20), quillaja saponin (SQ) and rhamnolipid (Rha). The MLCD-based SLNs had a smaller size and lower surface charge than TP-SLNs with a Cur encapsulation efficiency of 87.54–95.32% and the Rha-based SLNs exhibited a small size but low stability to pH decreases and ionic strength. Thermal analysis and X-ray diffraction results confirmed that the SLNs with different lipid cores showed varying structures, melting and crystallization profiles. The emulsifiers slightly impacted the crystal polymorphism of MLCD-SLNs but largely influenced that of TP-SLNs. Meanwhile, the polymorphism transition was less significant for MLCD-SLNs, which accounted for the better stabilization of particle size and higher encapsulation efficiency of MLCD-SLNs during storage. In vitro studies showed that emulsifier formulation greatly impacted on the Cur bioavailability, whereby T20-SLNs showed much higher digestibility and bioavailability than that of SQ- and Rha-SLNs possibly due to the difference in the interfacial composition. Mathematical modeling analysis of the membrane release further confirmed that Cur was mainly released from the intestinal phase and T20-SLNs showed a faster release rate compared with other formulations. This work contributes to a better understanding of the performance of MLCD in lipophilic compound-loaded SLNs and has important implications for the rational design of lipid nanocarriers and in instructing their application in functional food products.

## 1. Introduction

Curcumin (Cur) is a natural polyphenolic bioactive isolated from the rhizomes of turmeric spices [1]. It has been widely used as spice and colorant in curries, cosmetics and nutraceuticals for health purposes owing to its well-known biological benefits such as anti-oxidative, anti-microbial, anti-inflammatory and anticancer activities [2]. However, its high susceptibility to oxidation, liability to degradation and low bioavailability limit the utilization of Cur in functional food products. By incorporating it into a protective delivery system, the chemical stability can be well protected, and the bioavailability can be enhanced facilitating its application in the fields of pharmacology and functional foods [2]. Several lipid-based formulations such as oil-in-water emulsions, micelles [3], niosomes [4], nanostructured lipid carriers [5], solid self-nanoemulsifying drug-delivery systems [6], liquid crystalline nanostructures, liposome-micelle-hybrids [7], liquid crystalline lipid nanoparticles [8] and liposomes [9] have been designed to entrap Cur and increase the absorption and efficacy in the targeted site. The bioavailability of Cur can be influenced by the form of carriers and interfacial properties [10,11].

Encapsulation matrices including solid lipid nanoparticles (SLNs) and nanostructured lipid carriers belong to the colloidal oil-in-water dispersions of solid lipids, which can be conveniently applied in food products such as beverages, yogurts, dressings and soups as carrier systems in protecting bioactive ingredients against chemical degradation and controlled release lipophilic compounds [12]. SLNs are fully crystalline lipid matrixes and have an organized crystalline structure with the bioactive components accommodated within the lipid matrix [13]. The solidified lipid matrix can efficiently restrict the mass transport of the incorporated functional agents or drugs to migrate to the particle surface where oxidation and degradation tend to occur [2].

The appropriate choice of solid lipid matrix and emulsifiers is of great importance to stabilize nanoparticles and determine the structural arrangement inside the SLNs. The change in lipid crystal morphology from a thermodynamically unstable α crystal to thermodynamically stable β crystal occurs easily during storage, which would cause the leakage of internal bioactives [14]. In addition, the crystalline transformation also increased the surface area and led to insufficient emulsifiers at the particle surface which subsequently caused particle aggregation especially when stabilized by high-melting emulsifiers [15,16]. It has been demonstrated that substituting part of a soybean phospholipid with a hydrogenated soybean phospholipid in curcumin-loaded liposomes can enhance the stability and release performances of curcumin [9]. The liposomes composed of lecithin and emulsifiers including Tween 80 and decaglycerol monolaurate formed a new liposome-micelle-hybrid which can increase the solubility of curcumin in water [7]. Moreover, the self-assembled lyotropic liquid crystalline nanostructures formed by monoolein-PEGylated lipid mixture can serve for the encapsulation of additional active substances to target various disease mechanisms [17]. Generally, the commonly used surfactants to stabilize SLNs are synthetic and non-biodegradable. Natural or biosurfactants are possible choices as a replacement. *Quillaja saponins* (SQ) composed of hydrophobic triterpene aglycone and hydrophilic sugar moieties have good emulsifying properties and have been applied as an emulsifier to stabilize tristearin by forming a compact interfacial layer which can inhibit the crystal transformation and degradation of the encapsulated bioactives [1,18]. Rhamnolipids (Rha) are surface-active glycolipids composed of a polar rhamnose group and a nonpolar fatty acid group, which can be produced by microbial fermentation [19]. Both surfactants are natural surface-active biosurfactants. Meanwhile, although consumers are increasingly demanding healthy food ingredients, triglycerides or saturated fatty acids are typically used as the solid core lipid for SLNs [20]. Therefore, functional lipids offer new opportunities for designing particles with beneficial features. Diacylglycerol (DAG) has the capacity to reduce the postprandial fatty acid level, decrease the total cholesterol and LDL-cholesterol in serum and suppress the fat accumulation [21]. It thus can be applied as a healthy solid lipid substitute for triacylglycerol (TAG). In addition to owning good emulsifying properties, medium- and long chain diacylglycerol (MLCD) also has features of rapid metabolism and can supply instantaneous energy. In our previous research, the SLNs fabricated by MLCD were efficient Pickering particles to stabilize W/O emulsions with good plasticity [16]. Therefore, incorporating active ingredients into high-melting MLCD-based products is an excellent alternative to deliver bioactive compounds. In a previous study, 1-laurin-3-palmitin was found to be able to form thermodynamically stable β-form polymorph which would overcome the polymorphic transition issue faced by SLNs [22]. Moreover, 1,3-glycerol distearate was proved to possess good prospects as a solid lipid to fabricate nanostructured lipid carrier (NLC) with high entrapment efficiency of krill oil [23]. However, few investigations have been conducted concerning the influences of the glycerides and emulsifier types on the structural profiles of Cur-loaded SLNs and their digestion behaviors, although interfacial rigidity has been shown to be effective in retaining Cur and increasing its bioavailability [1].

The purpose of this study was thus aimed at studying the effect of different solid lipids, namely MLCD and tripalmitin (TP), on the physical stability and polymorphic behavior of SLNs loaded with Cur during storage. The influence of three emulsifiers, SQ, T20 and Rha on the physical and polymorphic stability of the SLNs, the Cur release and chemical stability during storage and the bioavailability of Cur after the in vitro-simulated gastrointestinal digestion process were also studied. The results can provide valuable information for designing Cur-SLNs by using functional lipids for application in food products.

## 2. Materials and Methods

### 2.1. Materials

Cur (≥95%) and porcine bile salt (cholic acid sodium salt ≥60%) were purchased from Yuanye Bio-Technology Co., Ltd. (Shanghai, China). Soybean oil was purchased from a local grocery store (Guangzhou, China). Medium-long chain diacylglycerol (MLCD) was synthesized in the laboratory (90.01% MLCD, 6.27% TAG, 2.67% monoglyceride and 1.05% free fatty acids). Glycerol tripalmitate (TP, 98%), glycerol trilaurate (TD, 98%), glycerol tristearate (TS, 98%) and rhamnolipid (Rha, ≥95%) were purchased from Macklin Biochemical Technology Co., Ltd. (Shanghai, China). Tween 20 (HLB, 16.7) was acquired from Usolf Chemical Technology Co., Ltd. (Shandong, China). *Quillaja saponins* (SQ, Sapogenin content 20–35%), Nile Red, Nile Blue, porcine pepsin and porcine lipase were purchased from Sigma-Aldrich (St. Louis, MO, USA). 

### 2.2. Cur-SLNs Preparation

Cur-SLNs were prepared through melt-emulsification and ultrasonication methods [16]. Briefly, the oil phase (5 wt.%, MLCD or TP) was fully melted at 80 °C, and Cur (0.6%, relative to the total lipid mass) was blended with the preheated lipid phase with stirring. The aqueous phase was prepared by dissolving T20 (2 wt.%), SQ (0.8 wt.%) or Rha (1 wt.%) in 5 mM phosphoric acid buffer solution (pH 7) and was heated to the same temperature as that of the oil phase. Subsequently, the hot aqueous phase was added into the oil phase and the mixture was homogenized at 10,000 rpm for 2 min at 80 °C with a rotor-stator homogenizer (IKA T25-Digital Ultra Turrax; Staufen, Germany) to produce a coarse emulsion. The emulsion was then ultrasonicated using a sonicating probe (ScientzII; Ningbo Scientz Biotecnology Co., Ltd., Zhejiang, China) for 7 mins (at intervals of 5 s, 480 W) at 80 °C. The dispersion was then cooled in an ice bath for the lipid to crystallize and the dispersion was then stored at 4 °C for further analysis.

### 2.3. Particle Size and Zeta Potential of Cur-SLNs

The hydrodynamic diameter (z-average), polydispersity index (PDI) and zeta potential of the Cur-SLNs were measured by using a dynamic light scattering analyzer (Zetasizer Nano, Malvern, Worcestershire, UK) at room temperature. The particle dispersions were diluted 100-fold in 10 mM phosphate buffer solution at pH 7 and equilibrated for 60 s to prevent multiple scattering. The refractive index of 1.54 and 1.33 was used for the solid lipid and water, respectively, to calculate the particle size distributions [24].

### 2.4. Transmission Electron Microscopy (TEM) Observation 

The morphology of Cur-SLNs dispersions was observed using the transmission electron microscope (TEM) (1200 EX II; JEOL Inc., Tokyo, Japan) operating at 120 kV and the micrograph was obtained by using a digital camera (FEI Tecnai-12; Hillsboro, OR, USA). All the Cur-SLNs dispersions were diluted 100-fold with ultrapure water before viewing. An aliquot (20 μL) of the sample was first dropped on a 200-mesh carbon Formvar-coated cooper grid for at least 1 min. Then, the sample was negatively stained with phosphotungstic acid (2.0 wt.%) for 15–20 s and dried at room temperature before observation.

### 2.5. Stability of Cur-SLNs against pH and Ionic Strength 

The stability of Cur-SLN was evaluated by adjusting the pH of the suspension from 2 to 7 using 0.1 M NaOH or HCl at ambient temperature. The ionic stability of Cur-SLNs was assessed by adding NaCl (0–200 mM) to the dispersions at ambient temperature. The z-average and zeta potential of samples were measured 2 h after equilibrium [1].

### 2.6. Thermal Analysis

The melting and crystallization behaviors of bulk fat and Cur-SLNs dispersions were determined using a differential scanning calorimeter (DSC1 500 instrument; Mettler Toledo, Zurich, Switzerland) equipped with a METT-FT900 temperature control unit (Julabo, Switzerland). Approximately 8 mg of freshly prepared samples or the bulk fat was weighted and hermetically sealed in an aluminum pan and cooled from 80 °C to 10 °C at a rate of 5 °C min^−1^, held for 5 min at 5 °C, and then heated to 80 °C at a rate of 5 °C min^−1^ under constant nitrogen flow of 40 mL min^−1^ [16]. An empty aluminum pan was used as the reference. The peak melting temperatures, crystallization onset temperature and enthalpies were determined using DSC data analysis software (Mettler Toledo International, Zurich, Switzerland).

### 2.7. X-ray Diffraction (XRD)

Crystal polymorphism was observed using an MSAL-XD-II X-ray diffractometer (Bruker AXS, Karlsruhe, Germany) equipped with Cu Kα radiation (λ = 1.54 Å, tension of 40 kV and 40 mA) and a Ni filter. Measurement was performed in a 2θ range of 10° to 30° with a scanning step size of 0.02° [16]. All samples were measured at ambient temperature. The d-values were calculated using the following formula:d = nλ/2sin θ(1)
where λ is the wavelength and θ is half of the scanning angle.

### 2.8. Fourier Transform Infrared (FT-IR) Spectroscopy

The intermolecular interactions between different components in the SLNs were studied using Fourier transform infrared (FT-IR) spectroscopy. Briefly, Cur, MLCD, TP and freeze-dried powder of Cur-SLNs were mixed with potassium bromide with a mass radio of 1:100 to make pellets for measurement using the Nicolet iS50 FTIR Spectrometer (Thermo Fischer Scientific Inc., Waltham, MA, USA). The FT-IR spectra were acquired with 16 scans at a wavenumber from 500 to 4000 cm^−1^ [25].

### 2.9. Entrapment Efficiency (EE) of Cur in SLNs

The EE of Cur in SLNs was evaluated as previously reported [25]. Briefly, 1 mL of Cur-SLNs dispersions were centrifuged at 12,000× *g* for 15 min to remove the unloaded Cur crystals. The supernatant was then transferred and diluted by mixing 100 μL of this supernatant with 900 μL of ethanol, and the encapsulated Cur concentration in the supernatant was measured using a UV-Vis spectrophotometer (Alpha-1506, Shanghai, China) at 425 nm. The Cur content was calculated by referring to a linear calibration curve established using standard solutions of Cur dissolved in ethanol (R2 = 0.9998). The EE was determined by calculating the percentage of Cur in the supernatant with respect to the initially added amount in the solution. The EE of Cur in SLNs was determined using the following equation:(2)$$\mathrm{EE}(\%)=100\times \frac{\mathrm{Amount}\;\mathrm{of}\;\mathrm{encapsulated}\;\mathrm{of}\;\mathrm{Cur}}{\mathrm{Total}\;\mathrm{Cur}}$$

### 2.10. Chemical Stability of Cur in SLNs

The chemical stability of Cur after storage at 4 °C for 4 weeks was evaluated to investigate the protective effect of Cur-SLNs. Briefly, the Cur-SLNs dispersion (100 μL) was diluted 100 times using ethanol and sonicated for 30 min to release Cur. The residual amount of Cur was quantified using the above UV-Vis spectrophotometer (425 nm) and calculated as the percentage of Cur with reference to that on day 0. The control was the free Cur dissolved in ethanol [25]. 

### 2.11. Simulated Digestion

#### 2.11.1. Gastrointestinal Digestion Model In Vitro

The gastrointestinal tract digestion model was based on the methodology of a previous study with minor modifications [26]. Briefly, 4 mL of Cur-SLNs was diluted to 20 mL with distilled water and mixed with 20 mL of gastric fluid at 1:1 ratio. The pH of the mixture was adjusted to pH 2.5 with 0.1 M NaOH or HCl solution. The mixture was shaken in a water bath for 2 h (37 °C, 120 rpm). Then, 30 mL of the gastric phase was adjusted to pH 7.0 and mixed with 1.5 mL of simulated intestinal fluid and 3.5 mL of bile salt solution at 37 °C water bath for 2 h with constant shaking. Subsequently, 2.5 mL of 2.4 mg mL^−1^ lipase solution was added, and the mixture was stirred for 2 h at 250 rpm in a 37 °C water bath. The pH was maintained at 7.0 with an automated titration device (Metrohm USA Inc., Riverview, FL, USA) by titrating 0.1 M NaOH solution into the mixture. The volume of NaOH consumed was recorded and the amount of free fatty acids (FFAs) released was calculated from the titration curve using the following equation:(3)$$\mathrm{FFA}(\%)=100\times \frac{V_{\mathrm{NaOH}}{\times M}_{\mathrm{NaOH}}{\times M}_{\mathrm{lipid}}}{2{\times W}_{\mathrm{lipid}}}$$
where V_NaOH_ and m_NaOH_ are the volume and molarity of NaOH added, respectively, and M_lipid_ and W_lipid_ are the molecular weight and weight of the lipids in the small intestine phase, respectively.

#### 2.11.2. Measurements of the Particle Size and Zeta Potential of Digesta

The digestive fluids’ hydrodynamic diameter (z-average) and zeta potential were determined using a dynamic light scattering analyzer (Zetasizer Nano, Malvern, Worcestershire, UK) at room temperature. The gastric phase and intestinal phase samples were diluted with phosphate buffer prior to analysis to avoid multiple scattering effects.

#### 2.11.3. Morphology of Cur-SLNs after Digestion

The samples after gastrointestinal digestion were viewed using a Zeiss LSM 510 confocal laser scanning microscope (Carl Zeiss Inc., Oberkochen, Germany). MLCD and TP were stained separately with Nile blue and Nile red dissolved in acetone (0.2 mg mL^−1^), respectively. The dye solution (20 μL) was dropped into the samples (200 uL), stained for 1 min and placed on a glass slide followed by covering with a cover slip. Images were obtained with a 20× objective lens, under excitation/emission wavelengths of Nile red, Nile blue and Cur of 550/630 nm [27], 633/697 nm [16] and 425/480 nm [27], respectively. 

#### 2.11.4. Cur Bioavailability

After simulated intestinal digestion, the digestion media were collected immediately and centrifuged at 12,000× *g* rpm for 30 min (at 4 °C) in a high-speed centrifuge [28]. The micellar phase containing dissolved Cur was collected and analyzed using a UV-Vis spectrophotometer (425 nm). The bioavailability of Cur was calculated using the following equation:(4)$$\mathrm{Bioavailability}\%=\frac{\mathrm{amount}\;\mathrm{of}\;\mathrm{solubilized}\;\mathrm{curcumin}\;\mathrm{in}\;\mathrm{micelle}}{\mathrm{amount}\;\mathrm{of}\;\mathrm{curcumin}\;\mathrm{in}\;\mathrm{original}\;\mathrm{emulsion}}\times 100\%$$

### 2.12. In Vitro Release Kinetics of Cur

The determination of the release kinetics of Cur in SLNs under simulated digestion conditions was measured using the dialysis membrane method [25]. Simulated gastric fluid (SGF) and simulated intestinal fluid (SIF) were mixed with an equal volume of pure ethanol to prepare the release medium. The 5 mL of freshly prepared Cur-SLNs dispersion was loaded into a dialysis bag with 3500 Da molecular weight cutoff, followed by immersion into the 150 mL of release medium in a water bath. During incubation in SGF for 2 h (37 °C, 120 rpm) and then SIF for another 2 h (37 °C, 250 rpm) under shaking in the dark, 4 mL of the release medium was collected at a predetermined time point, and an equal volume of freshly prepared release medium was replenished to maintain the constant volume. Cur content was determined using a UV-Vis spectrophotometer at 425 nm with a calibration curve. The in vitro release kinetics were analyzed using different models. 

### 2.13. Statistical Analysis 

All data were expressed as mean value ± standard deviation (SD) of three independent experiments and one-way analysis of variance (ANOVA) was performed using SPSS 16.0 statistical software (SPSS Inc., Chicago, IL, USA). Statistical differences were significant with a *p* value of <0.05 using Duncan’s new multiple range post hoc test.

## 3. Results and Discussions

### 3.1. Characterization of SLNs

Cur-loaded SLNs were prepared with an internal lipid core of 5 wt.% medium- and long chain diacylglycerol (MLCD) or tripalmitin (TP) using different surfactants. The particle size was measured and the results are shown in Figure 1A,B. The T20- and SQ-stabilized SLN dispersions showed bimodal particle size distributions. The sizes of MLCD-T20 and MLCD-SQ SLNs were 160.65 ± 1.77 nm and 376.93 ± 8.69 nm, respectively, smaller than those of TP-T20 and TP-SQ SLNs (457.03 ± 12.61 nm and 514.03 ± 19.5 nm, respectively). The smaller size of MLCD-SLNs was possibly due to the emulsifying properties of diacylglycerol which facilitated the formation of more stable nanoparticles. Meanwhile, T20 has a lower molecular weight and a greater HLB (16.7) than SQ, which was more efficient in facilitating its migration to the interface and formation of smaller particles. All Cur-SLNs demonstrated low PDIs (<0.4) indicating that they processed good dispersibility. Compared with the SLNs without Cur, the size and polydispersity of nanoparticles were not significantly influenced by the presence of Cur in lipids. As observed in Figure 1C, all the SLNs had negative charges with zeta potentials ranging from −22.6 ± 0.95 mV to −52.06 ± 1.3 mV. The relatively high zeta-potential was indicative of a strong repulsive force between the particles, which would reduce the aggregation tendency and increase the stability of multiphase systems [16]. It has been revealed that for electrostatically stabilized nanodispersions, a minimum zeta potential of 30 mV is necessary to maintain its stability and a minimum of 20 mV if the combined impact of electrostatic and spatial stabilization coexist [29]. The negative charge of T20-stabilized SLNs is caused by the hydrophilic polyoxyethylene head group forming hydrogen bonds with OH- in water and adsorption at the oil–water interface [30]. Due to the presence of carboxylic acid groups, saponin is an anionic surfactant with a significant negative charge [31]. Additionally, the Rha-SLNs were highly negatively charged due to the carboxylic acid groups present in the hydrophilic head, which can stabilize the dispersions by significant electrostatic repulsion [32]. Moreover, MLCD-SLNs showed a lower surface charge than TP-SLNs due to their different structure and composition. 

As demonstrated by the TEM results (Figure 1D), all SLNs showed spherical morphological structure. The particle sizes were in the range of 200–500 nm and the sizes of TP-SLNs were larger than that of MLCD-SLNs, which was consistent with the DLS data (Figure 1B). However, for the TP-T20 and TP-SQ SLNs, the particle size significantly increased after 4 weeks of storage, indicating that the SLNs started to aggregate to high degrees. This was possibly due to the polymorphic transition from α-polymorph via an orthorhombic β′-polymorph to a platelet-shaped β-polymorph during storage which increased the hydrodynamic radius or diameter of SLNs [14,24]. It is also noted that MLCD-SLNs were more stable during storage possibly due to the better surface activity of MLCD which can assist the particle stabilization. 

### 3.2. pH and Ionic Stability of Cur-SLNs

The physical stability of Cur-SLNs under different environmental conditions was important for their commercial application. The appearance, structure and release of bioactive substances will impact on their performance in food products such as beverages or skim milk [1]. From the particle size and visual appearance of SLNs under various pH ranges (2.0–7.0) after 2 h ambient storage (Figure 2A,C), it can be observed that the Rha-stabilized SLNs were more pH-sensitive, exhibiting clear signs of phase separation and increase in particle size when pH was reduced from 6 to 2. In contrast, no droplet aggregation was observed for the SLNs stabilized by T20 and SQ. The origin of the negative charge on the Rha-coated droplets was attributed to the presence of carboxylic acid groups in the structure of the rhamnose molecules. Typically, the pKa values of the carboxylic acid groups of Rha are around pH 4.28–5.50, which could explain the sharp increase in particle size and decrease in zeta potential observed in the vicinity of these pH values [32]. T20-stabilized SLNs gradually grew to a size of 700 nm, and their zeta potential was close to 0 mV when the pH was reduced to 2. The higher physical stability is because T20 stabilizes the emulsion primarily through spatial repulsion by the polymer head group making it relatively pH-insensitive. The SQ-coated oil droplets maintained a strong negative charge throughout the pH range. They had low size change and fluctuation, possibly due to the ionization of carboxyl groups on the SQ molecules producing strong spatial repulsion and the formation of stable interfacial films contributed to the stability of pH fluctuations [33]. Similarly, it was shown that tea saponin emulsion was resistant to droplet aggregation and phase separation at all measured pH values by offering spatial site resistance [34]. The high stability of SLNs dispersions under neutral conditions was mainly attributed to the strong electrostatic repulsion between highly charged droplets. However, this repulsion was insufficient to overcome the attractive interactions (e.g., van der Waals) between droplets as the pH decreased, which led to droplet aggregation [35]. 

The effect of ionic strength was further investigated by adding different levels of NaCl (0–200 mM) to the SLNs dispersions and then measuring their physical stability after 2 h of storage. As observed from Figure 2B,D, the average particle size of T20 dispersions showed little change and the appearance showed no obvious sign of phase separation. This result highlighted that the droplets in T20-stabilized SLNs were stabilized mainly by spatial repulsion to prevent aggregation. The slight fluctuation in particle size was possibly related to the dehydration of the hydrophilic polymer head group of the nonionic surfactant through salt addition [34]. Similarly, Uluata et al. (2015) observed that the droplet size of Tween 80-stabilized nanoemulsions increased from 0 to 100 mM of salt concentration, but remained stable at higher salt levels [36]. SQ-stabilized SLNs maintained small particle sizes over the range of salt ion concentrations tested, reflecting their high ionic stability. In general, the electrostatic shielding effect of NaCl led to a decrease in the surface charge of SLNs, which in turn reduced the electrostatic repulsion and increased the particle size especially for TP-Rha. As can be observed, the Rha-stabilized MLCD and TP SLNs showed a similar susceptibility to salt-induced agglomeration, and when above a critical salt concentration, the repulsive interactions were possibly lower than the attractive interactions and caused droplet aggregation [32]. Overall, the MLCD-Rha SLNs showed a higher stability with less intense phase separation compared with that of TP-Rha SLNs after 4 weeks of storage (Appendix A), indicating the high environmental stability of MLCD-SLNs.

### 3.3. Thermodynamic Properties and Crystal Polymorphism of SLNs

Understanding the thermal behavior of the lipid phase of Cur-SLNs dispersions is essential to optimize their formation, stability and functionality. Therefore, DSC was used to determine the melting and crystallization curves of solid lipid and Cur-SLNs dispersion stabilized by different surfactants. The main onset/peak temperatures and cooling/melting enthalpies are summarized in Appendix A. As shown in Figure 3A, MLCD showed three melting peaks at 51.75 ± 0.05 °C, 57.27 ± 0.12 °C and 68.26 ± 0.09 °C, representing the existence of low- and high-melting fractions of the MLCD [16]. TP started to melt at ~46 °C indicating the melt of α crystals, followed by an exothermic peak and the melting of β crystals at ~66 °C, and the crystallization curve showed an exothermic peak at 40 °C. The α crystal is a low-energy crystal form with melting and recrystallization occurring simultaneously; thus, the melting curve showed continuous endothermic and exothermic peaks [37]. It was also noted that although the melting point was similar to that of TP, the crystallization onset temperature of MLCD (60.92 ± 0.11 °C) was much higher than that of TP (41.67 ± 0.09 °C), which was in agreement with the previous report that DAG has a higher melting point than TAG with a similar fatty acid composition [38]. 

As shown in Figure 3B, the crystallization temperatures of all samples were always significantly lower than the corresponding melting temperatures, which can be explained by the presence of nucleation sites required for the formation of the ordered phase upon cooling, whereas the order–disorder transition did not require nucleation upon heating [19]. The thermal transition of the emulsified fat was significantly different from that observed in the bulk fat, and the decrease in the crystallization temperature of the SLNs was due to the supercooling effect within the nanoparticles [39]. In the colloidal SLNs, the impurities in the lipid phase were dispersed among a large number of lipid droplets, in which case nucleation occurs by a homogeneous mechanism that suppresses the crystallization and causes a high degree of supercooling [40,41]. Among the samples, stabilization with T20 led to a single crystallization peak with a much lower crystallization temperature than stabilization by other emulsifiers. This was because of the interaction between the fatty acyl moiety of the T20 molecules with the MLCD or TP in the dispersed phase [16]. Meanwhile, the Rha-SLNs showed broader crystalline peaks indicating a less ordered crystal structure. This was possibly due to the decrease in particle size, increase in surface area of particles and interaction between the emulsifier and lipid matrix [42]. Emulsifiers could penetrate the fatty matrix by hydrophobic tails, forming a grain boundary and delayed molecular reorganization, leading to delayed lipid crystallization and transition to the stable crystal form [16]. The saponin molecules formed a solid two-dimensional adsorption layer with dense molecular packing, and multiple hydrogen bonds between the sugar moieties of adjacent saponin molecules, which caused inhibition of the polymorphic transition [43]. Therefore, the broader endothermic peaks of SLNs indicated a less ordered crystal structure and some lipids were in an amorphous state by the emulsifying process.

The crystallographic properties of SLNs were further investigated using XRD as lipid crystallization morphology significant impacts lipid digestion and bioactive substances’ encapsulation. As observed in Figure 3C, the bulk MLCD showed the presence of crystals of β, α and β′ form at 4.59, 4.15 and 3.8 Å, respectively, and β form was the dominant form, whereas the bulk TP showed the presence of β and β′ at 4.62, 3.71 and 3.86 Å, respectively, with some other peaks existing. Therefore, significant difference existed in the crystallization profile of MLCD and TP. After emulsification, MLCD-SLNs showed no significant difference in polymorphic forms with the coexistence of α, β and β′ form stabilized by different surfactants and the crystallization peaks corresponding to the α-polymorph were larger in SLNs than those in the bulk MLCD, indicating the inhibition of the transition to a more stable β crystalline form by the surfactants. In contrast, the polymorphic forms of TP-SLNs were significantly different (Figure 3B). The TP-T20 SLNs showed diffraction peaks at 4.57 Å, 4.14 Å, 3.84 Å and 3.69 Å and the dominance of β crystals (4.57 Å). The TP-SQ SLNs showed one prominent diffraction peak at 4.15 Å indicating the dominance of α crystals, and the TP-Rha SLNs showed diffraction peaks mainly at 4.56, 4.13 and 3.71 Å, indicating the coexistence of β, α and β′ crystals [44]. With a similar hydrocarbon chain to the C16 fatty acid chain in the TP molecule, T20 (with C12 alkyl chain) and Rha (with C8-C16 alkyl chain) tended to interact with TP, which restricted the tight packing of lipid molecules and enhanced the intermediate β′ form stability. In contrast, SQ mainly affects the growth and crystal form by forming a viscoelastic film with a dense structure on the lipid surface. The behavior is consistent with the findings by Salminen et al. (2016) whereby SQ-stabilized tristearin SLNs only formed an α-subcellular crystal form because of the molecular profile and steric hindrance effects [43]. Meanwhile, the characteristic peaks of Cur in SLNs were not observed in this experiment, indicating that the Cur was in an amorphous state and was well-encapsulated in lipid particles [45]. 

The proportion of the crystal form was further calculated and the results are shown in Appendix A. A high increase in β form ratio was detected for TP-SLNs compared with those of MLCD-SLNs, indicating TP-SLNs were more favorable for the structural transition of the crystals and MLCD-SLNs had a better polymorphic stability. The existence of a high ratio of α form crystals and the transition also partly accounted for the high increase in particle size for the SLNs during storage (Figure 1B). Among the emulsifiers, T20- and Rha-stabilized SLNs showed a more significant reduction in α form and an increase in β′ form than the SQ-stabilized SLNs, possibly due to the less compact interfacial film formed by the T20 or Rha and the disruption would facilitate the polymorphic transition [1]. The increased viscosity or immobilization of the molecules in the interfacial vicinity in the presence of SQ impeded the conformational reorientation necessary for the α to β transition [43].

### 3.4. FT-IR Spectra, Encapsulation and Stability of Cur-SLNs

The FT-IR spectra of raw Cur, bulk lipids and lyophilized SLNs are shown in Figure 4A. The typical absorption peaks of Cur at 3508.36, 1599.59, 1511.12 and 1278.09 cm^−1^ belong to phenol O−H group stretching, C=C group stretching, aromatic ring C=C−C stretching vibration and aromatic ether group stretching [25,46]. For the glycerides, the absorption peaks around 2916 and 2849 cm^−1^ belong to the stretching of CH_2_−CH_3_ [46] and the peak around 1736 cm^−1^ belongs to the stretching of C=O, the absorption peaks around 1472, 1182 and 717 cm^−1^ ascribed to −CH_2_ bending vibration, −C−O stretching vibration and C=C out-of-plane bending vibration [47], respectively. Compared with pure solid lipids, the slight changes in some functional groups and peak intensities in the Cur−loaded SLNs spectra proved that hydrogen bonding or van der Waals interactions existed through C=C (alkene) and C=C−C (aromatic ring) in Cur with CH_2_−CH_3_ in solid lipids. The lipid structure showed little change in the presence of Cur, indicating that Cur was effectively loaded into the particles [48]. The small size and high encapsulation efficiency of SLNs were attributed to the neat arrangement of chemical bonds between the Cur and the lipid matrix [46]. Around 3500 cm^−1^, the spectrum of Cur-SLNs showed a broader absorption peak compared to bulk lipids, which indicated that the structure of the lipid matrix was less ordered or became amorphous. In addition, MLCD and TP-SLNs showed several dense and regular bands between 1200 and 1350 cm^−1^, which can be attributed to the all-trans arrangement of methylene groups and mutual coupling vibrations in lipids [49]. 

### 3.5. Encapsulation Efficiency and Chemical Stability of Cur-SLNs

The encapsulation efficiency (EE) of Cur-SLNs immediately after preparation and storage at 4 °C for 4 weeks were determined to investigate SLNs’ encapsulation ability and stability on curcumin as in Figure 4B. The EE of freshly prepared MLCD-T20, MLCD-SQ, MLCD-Rha, TP-T20, TP-SQ and TP-Rha were 95.32 ± 1.12%, 87.15 ± 0.67%, 89.54 ± 0.49%, 74.28 ± 2.99%, 82.97 ± 1.01% and 84.36 ± 3.07%, respectively. The high EE was related to the lipophilic properties of Cur and its good compatibility with the matrix of SLNs. The difference in EE may be related to several aspects including the particle size and the surface layer formed by the emulsifiers. The polar groups (such as hydroxyl groups) in Cur might interact with the polar head-group regions of lipids, which align the active compound in a lipid shell surrounding the lipid core. Compared with TP-based SLNs, MLCD-based SLNs had a better encapsulation efficiency for Cur. This may be because the lipid matrix of SLNs formed by MLCD with only two acyl chains has a larger space to encapsulate active substances. Moreover, the molecular polarity of MLCD might also increase its compatibility with Cur [50]. Additionally, the transition to a stable crystal structure might cause the exclusion of biologically active substances from the internal matrix and thus reduce the encapsulation ability. It has been reported that the bioactives were not tightly bound in the lipid matrix but interacted with the surfactants at the surface of nanoparticles, which can explain the linear decrease in EE with the SLNs’ size [51]. 

During storage, the transition to more ordered lipid crystals could reduce the number of amorphous lipids and imperfect crystals, which lead to the expulsion of Cur initially trapped in the voids of lipid bases. In the previous studies, surfactant type, carrier oil type, droplet size and pH significantly influence the chemical stability of encapsulated Cur [10,52]. Therefore, the Cur retention in SLNs was measured to assess its chemical stability. As demonstrated in Figure 4C, after 4 weeks of storage at 4 °C, the EE of Cur by MLCD-T20, MLCD-SQ, MLCD-Rha, TP-T20, TP-SQ and TP-Rha SLNs were 88.77 ± 2.95%, 77.09 ± 1.26%, 66.51 ± 3.18%, 70.63 ± 0.57%, 74.83 ± 0.21% and 65.52 ± 1.41%, respectively. The decrease in EE was more significant for Rha-stabilized SLNs and was mainly ascribed to its significant polymorphic transformation from the α to β form that occurred during storage and the reduction in lattice defects as demonstrated in Figure 3B. No direct relationship was observed between the chemical stability of Cur and the size of SLNs. This might be because the low storage temperature could efficiently keep the compound from oxidation and degradation. 

### 3.6. In Vitro Digestion of Cur-SLNs 

Under simulated intestinal conditions, the lipids were digested and Cur was released. The changes in the physicochemical properties of Cur-SLNs through a simulated digestion process were then compared and the bioavailability of Cur was then evaluated. The particle size and zeta potential of TP-based SLNs stabilized by different emulsifiers and the TAGs with varying chain lengths stabilized by T20 before and after the digestion process was also compared. The microstructures of SLNs at different digestion stages were recorded using CLSM to detect the red fluorescence of Nile red and Nile blue, and green fluorescence of the loaded Cur according to previous research [53].

As mentioned above, the initial Cur-SLNs were small in particle size (between 214.11 ± 16.38 nm and 574.11 ± 14.06 nm) and had high negative charges (between −18.87 ± 1.06 mV and −52.06 ± 1.91 mV) prior to digestion. As shown in Figure 5A,B, after stomach digestion, the larger particle size and a high degree of agglomeration were observed especially for Rha-SLN. This indicated that the SLNs were highly unstable to the acidic and high ionic-strength gastric environment attributed to the weakened electrostatic repulsion of the particles in the gastric environment. In contrast, the T20- and SQ-stabilized SLNs only showed a slight increase in size and had relatively high stability in the simulated stomach environment, consistent with previous studies [54]. In the stomach environment, the surface potential was significantly reduced due to the protonation of carboxyl groups on the polar region of the emulsifier under highly acidic conditions with high ionic strength and the presence of anionic pepsin. In contrast, the oil droplets remained resistant to aggregation suggesting that the oil droplets stabilized by T20 and SQ were mainly protected by spatial repulsion [55]. In particular, the SQ-stabilized SLNs showed a high surface charge in the stomach indicating a lower influence of the SQ film by the stomach environment and consequently, the droplets showed a lower degree of aggregation. 

The lipolysis of SLNs was then monitored by the small intestine digestion. As shown in Figure 5A,B, all digested SLNs dispersions contained similar particle sizes and zeta potential. The significant increase in the negative charge of the samples was due to the anionic nature of the bile salts and FFAs under small intestine conditions. It is hypothesized that lipid varieties and emulsifiers would influence the rate of micelle formation and the kinetics of Cur release from lipid droplets, affecting the final bioavailability. Therefore, the hydrolysis rate and degree for different SLNs dispersions were determined. In addition, TAGs with different fatty acids, TD and TS, were compared with TP. As shown in Figure 5C, the maximum FFAs released after the small intestine digestion phase for 2 h were 89.16 ± 1.29, 69.01 ± 0.87, and 46.15 ± 2.16% for MLCD-T20, MLCD-SQ and MLCD-Rha, respectively, higher than the previous report by using multilayer O/W emulsions to deliver the Cur (~72%) [56] and SLNs dispersions loaded with phytosterols (50–60%) [50]. Moreover, all the TAGs showed a lower FFAs’ release than MLCD-based SLNs. Specifically, the TP-based SLNs showed a very low lipid hydrolysis degree (27.94 ± 0.61%). The significant differences in the FFAs’ release level from SLNs with different solid lipids reflected the great influence of fatty acids’ varieties on the digestion of triglycerides. Generally, the digestion rates decrease with increasing the acyl chain length and the DAG was more readily digested by lipase compared with TAG [57,58]. Compared with TAG, MLCD has a higher dispersibility in aqueous media and can rapidly migrate into the surrounding aqueous phase; thus, MLCD could promote the interfacial lipase reaction and have a higher digestion rate. In addition, the higher emulsifying properties of MLCD would also make it easier for the droplets to be emulsified by bile salts and facilitate the lipase adsorption on the surface [59]. The low digestion ability of TP-based SLNs might be because palmitic acids existed at the sn-1 and sn-3 positions and sufficient amounts of calcium produced insoluble calcium soaps which prevented lipase from approaching the lipid droplets [60]. Such calcium soaps form crystals with very low water solubility and interfere with their binding in mixed micelles, leading to delayed fatty acid hydrolysis [61]. In addition, calcium ions can promote the activity of lipase, so reducing the calcium ion concentration also caused the reduction in fat digestibility [59]. Similarly, it has been reported that sn-2 mono palmitin has an inhibitory effect on lipolysis and the minimal transfer of C16:0 to the micelle phase caused the low bioavailability of bioactives [62].

Among the emulsifiers, T20-stabilized SLNs showed a faster digestion rate and higher levels of FFAs’ generation. T20 is a nonionic surfactant insensitive to pH changes and pepsin action. After gastric digestion, MLCD-T20 SLNs had a smaller particle size, which provided a larger specific surface area and more binding sites for lipase upon entering the intestinal phase. Similarly, Koukoura et al. found that T20-stabilized emulsions were inert to the oral–gastric–intestinal chemical environment; thus, can resist agglomeration in the gastric phase and avoid possible competition between T20 and bile salts at the interface [55]. The slower digestion rate and lower digestibility of MLCD-SQ SLNs might be related to the formation of interfacial expansion viscoelastic film which could inhibit the adsorption of bile salts and lipase and thus retard the digestion process [33]. Additionally, the highly anionic SQ and Rha might bind strongly to the cationic calcium ions, thus reducing the level of calcium ions available for precipitation and removing long-chain FFAs from the lipid droplet surface [63]. Significant aggregation of particles was observed in MLCD-Rha SLNs at the end of the stomach digestion, which could account for the degree of the reduction in lipid digestion. Rha molecules might diffuse into interparticle gaps at the interface and occupy the positions where bile salts and lipase can adsorb. Moreover, the Rha with a high negative charge and high emulsification activity provided strong electrostatic repulsion between droplets and negatively charged bile salts, consequently hindering the lipid hydrolysis [64]. 

The bioavailability of Cur refers to the Cur fraction of the mixed micelles available for intestinal absorption. As shown in Figure 5D, MLCD-T20 SLNs presented the highest Cur bioavailability (87.31 ± 1.29%), followed by MLCD-SQ and MLCD-Rha SLNs. Similarly, TD-T20 SLNs had higher bioavailability (81.87 ± 2.21%) than TS-T20 (65.98 ± 1.38%) and the TP-based SLNs showed the lowest bioavailability (38.45 ± 1.07%). The FFAs’ release level directly determined the ability of the micelle phase to solubilize Cur due to the incorporation of FFAs increasing the number of hydrophobic thresholds in the mixed micelles. The influence of emulsifiers on the bioavailability of Cur was related to the alteration of the surface area of droplets in the digestive tract or the binding ability of digestive enzymes to the droplet surface [65]. It has been demonstrated that the bioavailability of Cur in lipid particles was positively correlated with the chain length of the carrier oil. The lipid with long acyl chain provided more hydrophobic domains for incorporating the planer backbone of Cur [1]. In this study, MLCD-T20 SLNs showed higher bioavailability, indicating that the presence of C12 and C18 fatty acids retain Cur well and the controlled hydrolysis from SLNs was favorable for transferring Cur into the mixed micelles. However, the formation of large agglomerates in the MLCD-Rha and MLCD-SQ SLNs (Figure 5E) would reduce the release of Cur to the micellar phase and lead to lower bioavailability. In Figure 5E, the yielding of a yellow color by the overlapped red and green fluorescence indicated the Cur molecules were homogenously distributed in the lipid matrix of SLNs [53]. 

The structural transition from SLNs to mixed micelles upon intestinal absorption of the formulations is presented as a scheme shown in Figure 6. During digestion, bile salts and phospholipids were adsorbed to the lipid droplet surface through the “orogenic displacement mechanism” and the lipase/colipase complex subsequently adsorb onto the lipid–water interface, thus, allowing the lipase to enter inside the lipid molecules and convert the lipid to free fatty acids and monoglycerides [11]. As the reaction proceeds, a variety of molecular substances in the gastrointestinal fluid can assemble into colloids. For example, diglycerides, monoglycerides, free fatty acids, bile salts and undigested emulsifiers may be present as lipid droplets, micelles, vesicles or calcium soaps [63]. The micellized nutrients can then pass through the mucus layer and be absorbed by the intestinal epithelium.

### 3.7. In Vitro Release Kinetics of Cur

The dialysis membrane method was also applied to study the release profile of Cur in SLNs under simulated gastrointestinal conditions. Various kinetics models were fitted to understand the release mechanism of Cur from SLNs. As shown in Figure 7 and Appendix A, the Ritger–Peppas equation showed the best linearity for both free Cur and Cur-SLNs. In the Ritger–Peppas model, the coefficient n denotes the release (diffusion) index, depending on the release mechanism and the shape of the matrix. For cylindrical shapes, n < 0.45 means Fickian diffusion dominates, 0.45 < n < 0.89 means non-Fickian diffusion dominates and diffusion and skeletal dissolution act synergistically and with n > 0.89, skeletal dissolution dominates [66,67]. From the TEM image, it was known that the Cur-SLNs have a spherical structure. The diffusion index of 0.3897 for free Cur indicated that its release mechanism was Fickian diffusion and when the large concentration gradient across the membrane, it exhibited a sudden release to 61.74% by incubating in SGF for 2 h, followed by a sustained release in SIF, eventually reaching equilibrium with a cumulative percentage of 87.10% (Figure 7). In contrast, Cur in SLNs exhibited slower and sustained release kinetics under simulated gastrointestinal conditions with a synergistic release mechanism of drug diffusion and lipid matrix erosion and n values were 0.45~0.89 for all formulations, suggesting that the major drug is enriched in the core of SLNs and a small amount of drug in the shell can diffuse into the medium [68]. Overall, the release of Cur from SLNs was mainly accelerated by diffusion first from the lipid matrix to the surface of the SLNs’ lipid core, then through the surfactant layer to the aqueous phase, and finally to the release medium by a concentration gradient, while the lipid matrix was continuously degraded in the gastrointestinal digestive fluid. Because the SLNs delivery system did not release a large proportion of Cur in the SGF, it thus provides opportunities for the targeted and slow release of Cur in the intestine [25]. 

## 4. Conclusions

The results presented in this work evidenced that the lipid varieties of SLNs influenced its physicochemical properties and digestion profiles. MLCD-based SLNs showed higher storage stability with less increase in particle size than TP-based SLNs, possibly related to the better emulsifying properties of MLCD and the lower crystal polymorphism transition during storage. Although Rha exhibited the best emulsifying properties to fabricate SLNs of small sizes, the sensitivity to the pH variation and ionic environment caused significant aggregation of SLNs. The chemical stability and bioavailability of Cur were efficiently increased by encapsulation into the nanoparticles and Cur can be continuously released and transported for intestinal digestion. The in vitro digestibility of MLCD-based SLNs was dependent on the emulsifier varieties, whereby T20-SLNs showed much higher digestibility than SQ-SLNs with a high interfacial strength. The digestibility of SLNs was also influenced by the fatty acid chain length and the TP-SLNs had the lowest bioavailability. The above results indicated that MLCD-based SLNs were effective carriers for enhancing the hydrophilicity, chemical stability and bioavailability of Cur. The study also has important implications to aid in selecting emulsifiers and designing functional lipid-based nanocarriers for food applications.

## Figures and Tables

**Figure 1 foods-12-02045-f001:**
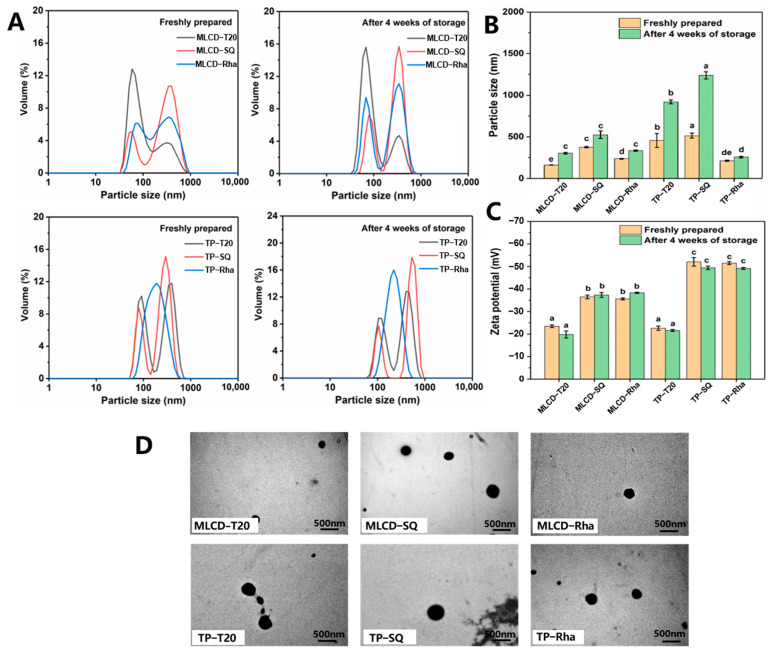
Particle size distribution (**A**), particle size value (**B**), zeta potential (**C**) of MLCD− and TP−stabilized Cur−SLNs freshly prepared by different emulsifiers on day 0 and after 4 weeks of storage at 4 °C. TEM micrographs (**D**) of MLCD− and TP−stabilized Cur-SLNs with different emulsifiers. Different letters indicate significant difference (*p* < 0.05) between samples at the same storage time.

**Figure 2 foods-12-02045-f002:**
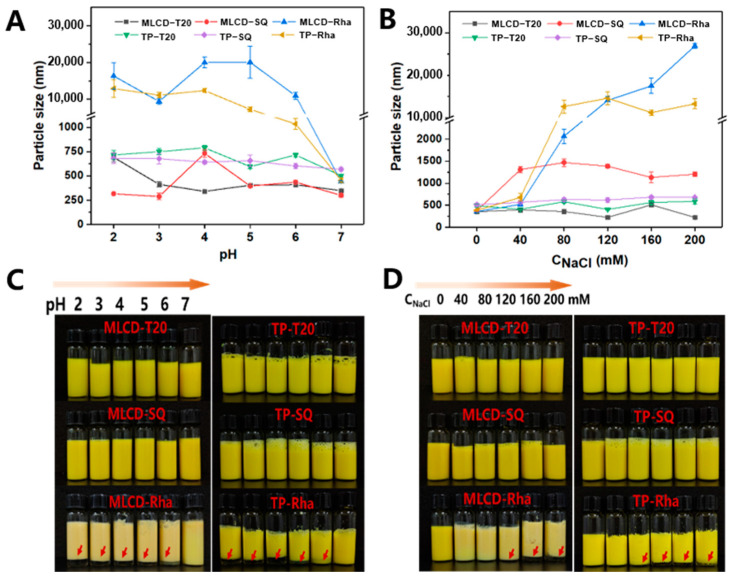
Particle size and visual appearance of Cur−SLNs with different internal solid lipid and emulsifiers under various pH values (**A**,**C**) and ionic strength (**B**,**D**) after 2 h ambient storage. The arrows denoted a phase separation occurred.

**Figure 3 foods-12-02045-f003:**
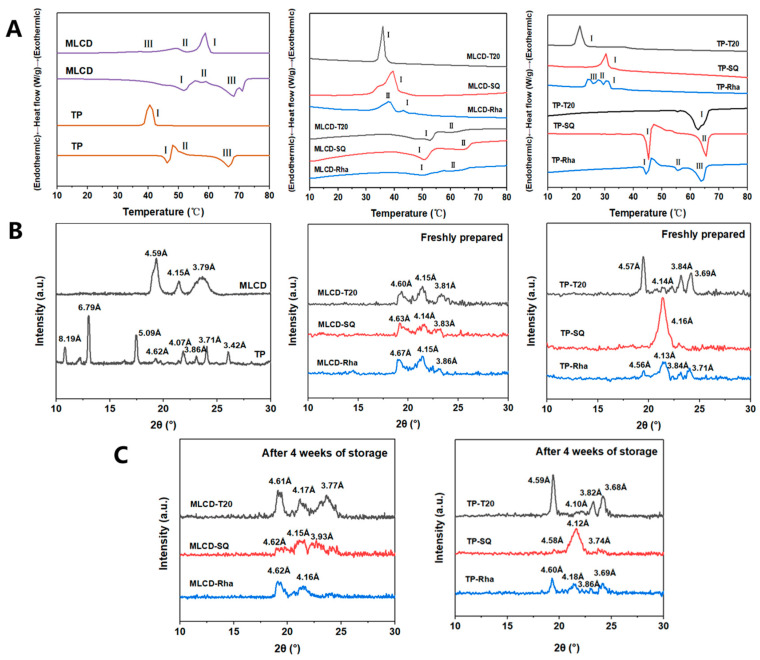
Melting and crystallization thermograms of bulk fat and MLCD−, TP−based Cur−SLNs (**A**); I–III in the figures indicate the peak position. XRD curves of bulk fat, freshly prepared MLCD−, TP−based Cur−SLNs (**B**) and Cur−SLNs stabilized by different emulsifiers after 4 weeks storage (**C**).

**Figure 4 foods-12-02045-f004:**
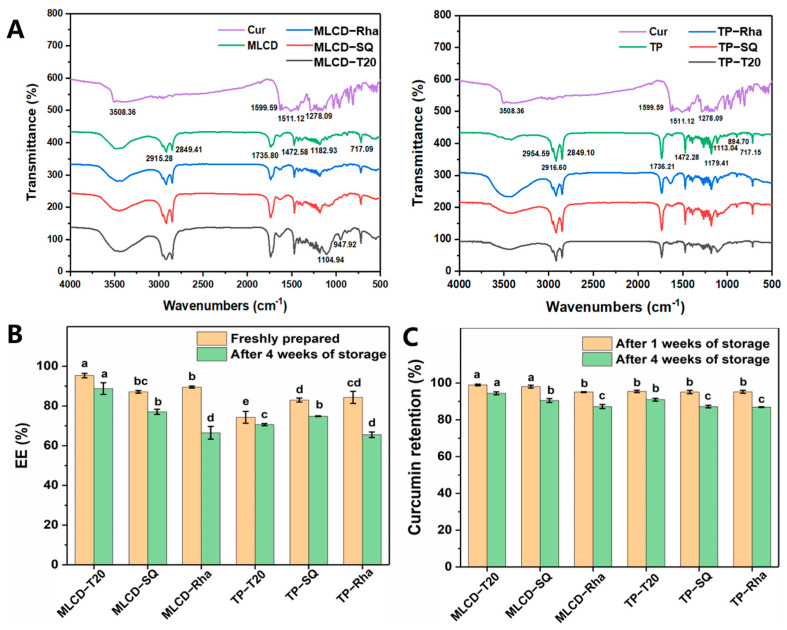
FTIR spectra of raw Cur, MLCD, TP and Cur−SLN powders (**A**); Encapsulation efficiency (**B**) and chemical stability (**C**) of Cur in SLN dispersions freshly prepared and stored at 4 °C for 4 weeks. Different letters indicate significant difference (*p* < 0.05) between samples at the same storage time.

**Figure 5 foods-12-02045-f005:**
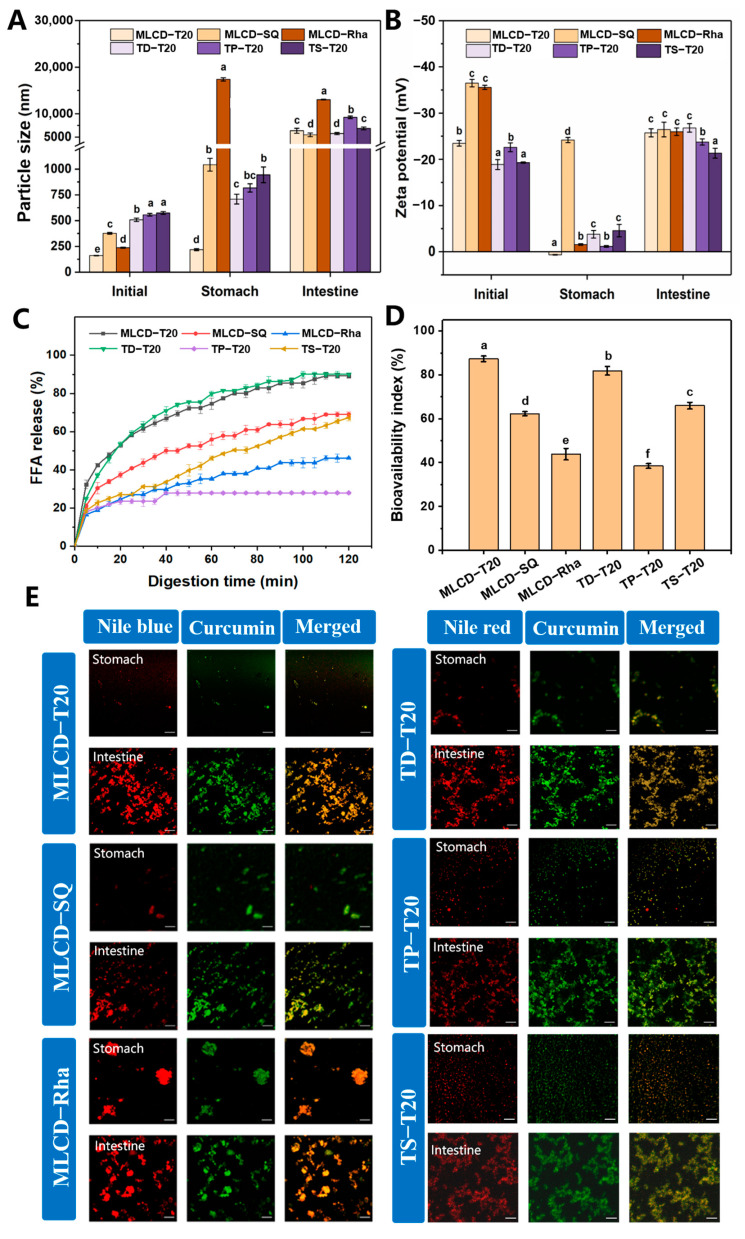
Particle size (**A**), zeta potential (**B**), FFA-released amount (**C**), Cur bioavailability (**D**) and CLSM images (**E**) of the digesta at various stages of simulated digestion. Different letters indicate significant difference (*p* < 0.05) between samples at the same digestion stage.

**Figure 6 foods-12-02045-f006:**
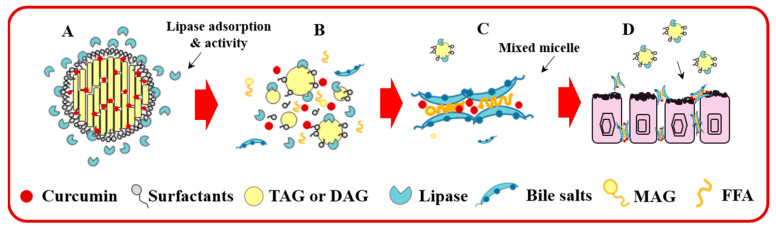
Scheme of lipolysis of Cur-SLNs in the small intestine: lipase adsorption (**A**), hydrolyzation (**B**), micelle formation (**C**) and passing through the mucus layer of the intestinal epithelium (**D**).

**Figure 7 foods-12-02045-f007:**
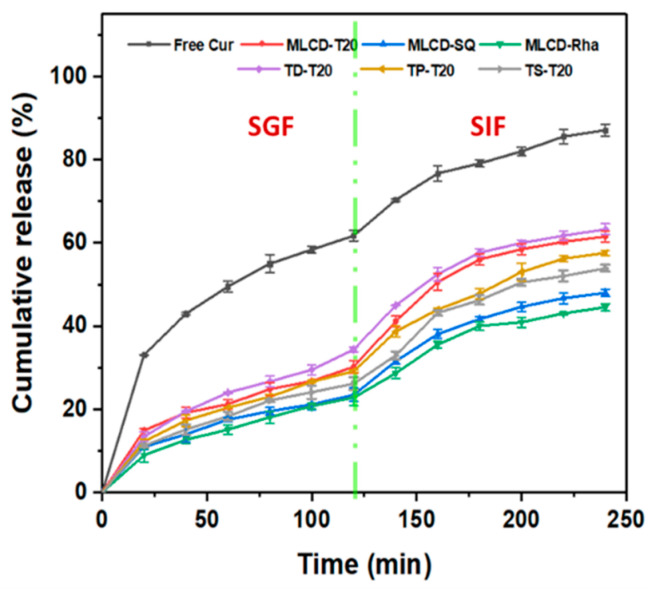
Release profile of free Cur and Cur loaded in SLNs during 120 min in SGF and the subsequent 240 min in SIF.

## Data Availability

Not applicable.

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
