# Peer review of "Physicochemical and In Vitro Digestion Properties of Curcumin-Loaded Solid Lipid Nanoparticles with Different Solid Lipids and Emulsifiers"

_foods, 2023, doi:10.3390/foods12102045_

Round 1

Reviewer 1 Report

The manuscript written by Yu et al., presents a very novel work. However, I feel the role of curcumin is under discussed. The authors have not shown the plethora of nano drug delivery used for curcumin formulation. The authors can use Niosomes (https://doi.org/10.3390/molecules27144634); nano lipid carrier (https://doi.org/10.3390/molecules25204610); SNEDDS (https://doi.org/10.3390/pharmaceutics14112395); Micelles (https://doi.org/10.3390/molecules28062693) to cite as nano drug delivery agents used as bioavailability enhancers and write in their paper. It seems like the author has not done a thorough review. I hope this will improve the quality of the manuscript. The authors should rephrase line #77-79. There are minor grammatical errors that can be accepted once the authors review the manuscript thoroughly. 

English to be extensively reviewed. 

Reviewer 2 Report

Physicochemical and in vitro digestion properties of curcumin- 2 loaded solid lipid nanoparticles with different solid lipids and emulsifiers

The manuscript is focused on important area of research, the writeup is good but there are some minor suggestions which needs to be addressed.

Line 114: Briefly, 5 wt% ???

Line 191: how the pH was adjusted??

Please read the methodology and change it for more clarification where necessary. Some sentences in the methodology section are not clear for the readers

Figure 1: please label the particles with approx. size

Why were TEM used for such comparatively large particles instead of SEM?? 

some sentences need to be rewrite for clarifications, particularly in methodology section 

Reviewer 3 Report

The present work describes the use of different solid lipids, emulsifiers and their impact on physical stability and characteristic behavior of curcumin loaded SLNs. Following are the minor comments regarding the manuscript titled “Physicochemical and in vitro digestion properties of curcumin-2 loaded solid lipid nanoparticles with different solid lipids and 3 emulsifiers”

Minor comments:

Line 23: Try to use simple and more understandable synonym for ‘subtly’.

Line 32: Better to also add optimized conditions for best formulation/s in the abstract.

Line 87: Here you have cited only one research article. Using few investigations is not justified until you provide more published researches.

Line 171: Its not clear whether Free or Entrapped Cur was calculated using this supernatant. Further, the centrifugation treatment was enough to separate the free/entrapped Cur? The protocol used for Entrapment Efficiency (EE) is not clear. Please make it clear and provide the EE formula.

Line 227: First write SGF and SIF in full, and then use abbreviations. 

Line 457: Clarify whether ‘[3] is pH value or citation. 

Please provide any leads towards the toxicity of SLNs and their further application.

Minor editing of English language may be considered to further improve the manuscript quality. 

Reviewer 4 Report

After carefully reading the manuscript entitled: "Physicochemical and in vitro digestion properties of curcumin-loaded solid lipid nanoparticles with different solid lipids and emulsifiers" it can be concluded that the authors spent a lot of time and effort in conducting experiments and writing an article. The topic is interesting and novel. However, a few things could be improved. Below are remarks and suggestions.

1.     Editing the English language, grammar, and style is required.

2.     Some sentences are too long and hard to follow up.

3.     Abstract missing important information. Reconsider extending it.

4. Figures are of low quality. Reconsider improving it and/or adding to the supplementary document in full size.

 Editing the English language, grammar, and style is required.

Reviewer 5 Report

The manuscript describes multicomponent compositions for the formulation of curcumin in solid lipid nanoparticles (Cur-SLN). These formulations comprise colloidal oil-in-water dispersions for the encapsulation and delivery of the active compound and are suitable for its protection from oxidation and degradation. Moreover, they can enhance the bioavailability of active molecules. The in vitro digestibility of the SLNs was found to be dependent on the emulsifier type. The reported data demonstrated the effect of the pH variation and the ionic environment on the physico-chemical properties and stability of the SLNs as well.

The manuscript is well prepared but should be addressed to broad readers. Application of curcumin-loaded nanoparticles in various application fields should be emphasized including curcumin-loaded lipid nanoparticles of the liquid crystalline type, e.g.

-Food Chemistry, 2020, 326, 126973. doi: 10.1016/j.foodchem.2020.126973;

-ACS Sustainable Chem. Eng. 2021, 9, 14821-14835. doi: 10.1021/acssuschemeng.1c04706;

-Smart Materials in Medicine 2022, 3, 274–288. doi: 10.1016/j.smaim.2022.03.001;

-Colloids and Surfaces A: Physicochemical and Engineering Aspects, 2022, 655, 130210. doi: 10.1016/j.colsurfa.2022.130210.

The term "bioaccessibility" can be replaced by "bioavailability" in the paper.

The data in Figure 1 are not adequately presented. The text describes bimodal distributions, while the particle size panel shows a single mean value. It is hard to appreciate the meaning of the letters (a) and (b) above the histograms. The caption should give more details about the measured data.

The structural transition from SLN to mixed micelles upon intestinal absorption of the formulations can be presented as a Scheme, which helps to understand the release curves of CUR in SGF and SIF.
